# Research of a Radar Imaging Algorithm Based on High Pulse Repetition Random Frequency Hopping Synthetic Wideband Waveform

**DOI:** 10.3390/s19245424

**Published:** 2019-12-09

**Authors:** Songhua He, Xiaotian Wu

**Affiliations:** School of Information Science and Engineering, Hunan University, Changsha 410082, China

**Keywords:** high pulse repetition frequency (HPRF), random frequency hopping (RFH), radar imaging, hypersonic aircraft

## Abstract

Aiming at the imaging algorithm of high-pulse-repetition random-frequency-hopping synthetic wideband radar on a supersonic/hypersonic aircraft platform, this study established an echo simulation model of target and clutter, analyzed the special range-Doppler coupling effect and its influence on imaging, and proposes a method of imaging with pipeline-parallel processing based on generalized 2D matched-filtering and Doppler pre-processing. In the method, Doppler-beam-sharpening was advanced to be performed with the pulse compression process in each frame, and the special range-Doppler coupling effect caused by high dynamic motion of platform and random frequency hopping in bandwidth synthesis was well suppressed; several modes of random frequency hopping were designed and the pipeline-parallel image processing algorithm was optimized for each mode. Theoretical analysis and simulation results show that the proposed imaging method can effectively avoid the divergence of 2D range-Doppler images in the range direction, and can meet the requirements of real-time imaging.

## 1. Introduction

In order to improve the performances of target-detection, low-probability-of-intercept, and anti-jamming, the high pulse repetition frequency (HPRF) synthetic wideband waveform has been used for radar imaging in supersonic/hypersonic aircraft guidance [1,2,3,4,5]. HPRF can decrease the velocity ambiguity, reduce the folding effect of clutter in Doppler direction, and then improve the signal-to-clutter ratio (SCR) and target detection ability. In addition, combined with random frequency hopping (RFH), HPRF can increase the number of accumulated pulses per unit time and improve the signal accumulation gain ratio of detector (coherent detection) to interceptor (non-coherent detection), which can decrease the peak power of the transmitting signal and improve the low-interception ability of radar. Furthermore, wideband HPRF RFH can improve the anti-jamming ability of guidance radar. RFH makes the reconnaissance jammer unable to predict the frequency-hopping pattern adopted in each frame; and makes it difficult to implement the answering-deception-jamming and the narrowband-blocking-jamming at each frequency point. HPRF and RFH enable the radar receiver to select echoes of a target from a relatively narrow range region according to the number of periods of return delay, and can suppress the deceptive-repeater-jamming, which is not in the same range region as the target, especially from the side-lobe direction.

In the case of RFH, the frequency domain sampling of radar signal in each frame is random and non-uniform. Because the basic frequency changes randomly between frames, the equivalent time-domain sampling of the inter-frame Doppler processing is also non-uniform and random. It is difficult to adopt the fast imaging algorithm based on Inverse Discrete Fourier Transform (IDFT) whether in pulse-compression processing at range direction or beam-sharpening processing in the Doppler direction. In addition, in the case of supersonic/hypersonic application, due to the large broadening in clutter Doppler spectrum, the special range-Doppler coupling effect and the large motion-compensation residue in the echoes, it is difficult to use the conventional method of fractal-dimension processing in the two directions of range and Doppler. Therefore, it is important to develop new imaging methods and fast algorithms for RFH radar. The existing non-uniform DFT (NU-DFT) fast algorithms, such as Vandermonde determinant method [6,7], regular Fourier matrix method [8,9], and min-max interpolation method [10,11], have specific constraints on the structural characteristics of non-uniform sampling signals, and their versatility is relatively poor. Compressed sensing technology [12] has also been widely used in the field of RFH synthetic wideband imaging [13,14,15,16,17,18,19]. According the theory of compressed sensing, it requires that the target and clutter background meet the basic conditions of sparsity. In the cases of wideband high-range-resolution (HRR) and high signal-to-noise ratio (SNR) or high SCR, compared with the number of range resolution units, the number of the observed targets or the number of scattering centers of the targets is limited. Therefore, it can provide a good guarantee for the sparsity in the high-resolution range profiles. The advantage of the compressed sensing method is that it can reconstruct the range profile through a small number of observations, which means in RFH radar, that it can effectively reduce the number of transmitting pulses without decreasing the resolution of the range profile. However, the compressed sensing algorithm needs a large number of matrix inversion operations, which makes it difficult to meet the real-time requirement, especially in hypersonic-platform-borne (HPB) radar imaging application of limited computing resources and extremely short platform-target intersection time. In addition, in the case of strong clutter and low SCR, the sparsity required by compressed sensing is also difficult to meet.

In view of the above problems, we established the scattering echo model of target and clutter for an HPRF RFH radar system and supersonic/hypersonic aircraft platform, and analyzed the special range-Doppler coupling effect and its influence on imaging. We also proposed the 2D range-Doppler imaging method and the 1D HRR imaging method based on Doppler pre-processing and 2D generalized matched-filtering (GMF) processing. Additionally, we designed several RFH modes, and proposed the corresponding pipeline-parallel processing, fast, real-time imaging algorithm for a different RFH mode. Theoretical analysis and simulation experiments showed that the proposed imaging method could effectively suppress the special range Doppler-coupling effect, achieve good imaging performances, and easily meet the real-time imaging requirements of supersonic/hypersonic aircraft-borne application.

## 2. Echo Modeling of RFH Synthetic Wideband Radar

Set the imaging processing time period of RFH radar as *M***N***T*. *M* is the number of sub-frames, which is the number of accumulated frames required for Doppler processing, the size of which defines the speed resolution; *N* is the number of frequency hopping points per frame, the synthetic bandwidth of which defines the range resolution; *T* is the pulse repetition period. In the case of HPRF, the echo delay *τ_H_* of targets and sea/land clutters from the main-lobe direction is much greater than *T*, *n_s_T* < *τ_H_* < (*n_s_* + 1)*T*, where *n_s_* is integer. Here we assume that each receiving complex frame lags behind the transmitting complex frame for *n_s_* periods and the period in the receiving complex frame is numbered (*n*|*m*), where *m* is the number of the frame (*m* = 0, 1, …, *M* − 1) and *n* is the number of pulse period in each frame (*n* = 0, 1, …, *N* − 1). In each pulse period, the receiving signal is sampled and the sampling interval is equal to the pulse-width *τ*; then, the total number of sampling points in each period is *K* = *INT*[*T*/*τ*] (*INT*[.] represents rounding down). Taking the starting time of each period as the reference, the corresponding sampling time is *τ*, 2*τ*, …, *Kτ*, which is numbered *k* = 0, 1, …, *K* − 1 respectively, and *k* is the number of sampling unit. By using the *n_s_*-period-delayed frequency hopping pattern to construct the local reference signal, the echo signal is coherently received, and the echo signal with a range of *cn_s_T*-*c*(*n_s_* + 1)*T* (*c* is the speed of light) is selected by the IF filter of receiver. For any scattering point in the selected range, according to the radar principle, the received/sampled signal can be expressed as follows:(1)$$\begin{array}{c} x\left(n|m,k\right)=Aexp\{j2\pi [\left(2R/c\right)\Delta f_{d}i_{mn}-\left(2f_{m}VNT/c\right)m-\left(2f_{m}VT/c\right)n-\\ \left(2\Delta f_{d}VT/c\right)ni_{mn}-\left(2\Delta f_{d}VNT/c\right)mi_{mn}-\left(2\Delta f_{d}V\tau /c\right)ki_{mn}-\left(2\Delta f_{d}V/c\right)\tau i_{mn}-\\ \left(2f_{m}V\tau /c\right)k+2f_{m}\left(R-V\tau \right)/c]\}. \end{array}$$

Here, the pure RFH mode (fast hopping intra frame/wide hopping inter frame; other modes are special cases of this mode) is investigated, where *f_m_* is the basic frequency of the transmitting signal at the *m*-th frame; Δ*f_d_* is the minimum frequency jump interval determined by the minimum quantization level of direct digital synthesizer (DDS); *i_mn_* is an integer which is randomly selected according to a certain frequency hopping pattern in the range of integer set [0, 1, 2, …, *I* − 1], where *I* = Δ*F*/Δ*f_d_* (the same value cannot be repeated in the same frame); Δ*F* is the synthetic bandwidth; *f_m_* + Δ*f_d_i_mn_* is the carrier frequency of the transmitting signal in the *n*-th period of the *m*-th frame; *R*′, *R*, and *V* are, respectively, the actual range, ambiguous range, and radial velocity of the point target at the starting time of the first pulse period of each receiving complex frame, $R^{\prime }=R+cn_{s}T/2$ and 0≤R<cT/2. The velocity is defined as positive for movement facing the radar and is assumed to remain unchanged within *MNT* (the time of a complex frame, usually several milliseconds). Here we ignore the influence of the slight change of *V* within a short time of milliseconds.

As shown in Figure 1, in the *n*-th pulse period of the *m*-th frame in each complex frame, the time delay of the receiving echo is $2\left(R-VmNT-VnT-Vn_{s}T\right)/c$ relative to the starting time mNT+nT of this period. Because the *k*-th sampling unit in each period can only acquire data of echo which has delay in the range of kτ − (k+1)τ, $k\tau <2\left(R-VmNT-VnT-Vn_{s}T\right)/c<\left(k+1\right)\tau$, and the echo is sampled by the *k*-th sampling unit.

It is apparent that if the scatter has a facing range-walk of δR in a complex frame and 0<R−ckτ<δR, then the echo is sampled first by the *k*-th unit in some frames, and then by the (*k* − 1)-th unit in the remaining frames, which is called cross-sampling-unit movement. For supersonic/hypersonic applications, considering the large-scale range-walk of cross-sampling unit in a complex frame, the following requirement must be met for each scattering point:$$kc\tau /2<R-VmNT-VnT-Vn_{s}T<\left(k+1\right)c\tau /2.$$

For (*n*|*m*, *k*) combinations that do not satisfy the above equation, *x*(*n*|*m*, *k*) = 0.

The advantage of the simulation model shown in Equation (1) is that it can fully describe the actual cross-sampling-unit movement in supersonic/hypersonic applications. In addition, it is suitable for panoramic simulation of echoes from the area that is illuminated by the main-lobe of radar beam. The panoramic clutter area can be divided into many grids, and echo from each grid can be simulated as point scattering by using Equation (1). The target can be simulated as multiple scattering centers, and echo from each scattering center can be simulated by using Equation (1). The target may be close to the junction of two sampling unit, and echoes from the target may appear successively at two adjacent sampling units (which is the so-called cross-sampling-unit range-walk). The clutter echoes appear at most sampling units (in the case of HPRF, *cT*/2 is larger than but very close to the radial length of the illuminated area of main-lobe). According to Equation (1), the return data containing both clutter-background and target can be simulated as follows:$$x_{S}\left(n|m,k\right)=x_{T}\left(n|m,k\right)+x_{C}\left(n|m,k\right),$$
where *x_T_*(*n*|*m*, *k*) is the target echo, which can be expressed as the sum of the point scattering echoes of multiple scattering centers; *x_C_*(*n*|*m*, *k*) is the clutter echo, which can be expressed as the sum of the point scattering echoes of each clutter grid in the main-lobe illuminated area. The amplitude of each clutter scattering point is randomly selected according to the Rayleigh distribution, and the parameter *σ*^2^ of the Rayleigh distribution is controlled according to the required SCR.

According to the moving speed of the platform, angle between the moving direction of the platform, and the illumination direction of the beam, the estimated value *V_C_* of the radial speed of the center of the main-lobe clutter can be obtained. Clutter-center velocity compensation is applied to data acquired in each complex frame as follows:(2)$$\begin{array}{c} y_{S}(n|m,k)=x_{S}(n|m,k)\times exp\{j2\pi [(2f_{m}V_{C}NT/c)m+(2f_{m}V_{C}T/c)n+\\ (2\Delta f_{d}V_{C}T/c)ni_{mn}+(2\Delta f_{d}V_{C}NT/c)mi_{mn}+(2\Delta f_{d}V_{C}\tau /c)ki_{mn}+(2\Delta f_{d}V_{C}/c)\tau i_{mn}+\\ (2f_{m}V_{C}\tau /c)k+2f_{m}V_{C}\tau /c]\}. \end{array}$$

Considering the Doppler broadening effect and the velocity estimation error of the moving platform, let *v* = *V* − *V_C_* be the velocity surplus of the scattered relative to the clutter center. After clutter-center velocity compensation, the sampled signal of each scatter can be expressed as follows:(3)$$\begin{array}{c} y\left(n|m,k\right)=Aexp\{j2\pi [\left(2R/c\right)\Delta f_{d}i_{mn}-\left(2f_{m}vNT/c\right)m-\left(2f_{m}vT/c\right)n-\\ \left(2\Delta f_{d}vT/c\right)ni_{mn}-\left(2\Delta f_{d}vNT/c\right)mi_{mn}+2f_{m}R/c]\}exp\left\{j\varphi \left(f_{0},R,v\right)\right\}, \end{array}$$
where $\varphi \left(f_{0},R,v\right)$ is a constant term independent of (*n*|*m*, *k*).

Because the velocity surplus of clutter or target is far less than the platform velocity, some phase terms in the compensated signal can be ignored, which can simplify the subsequent imaging process. The ignored phase terms are $j2\pi \left[-\left(2\Delta f_{d}v\tau /c\right)ki_{mn}\right]$, $j2\pi \left[-\left(2\Delta f_{d}v/c\right)\tau i_{mn}\right]$, $j2\pi \left(-2f_{m}v\tau /c\right)$, and $j2\pi \left(-2f_{m}v\tau /c\right)k$, the variation range of which is not more than *π*/4 in a complex frame.

## 3. High Quality Real-Time Imaging of HPB HPRF RFH Radar

### 3.1. Special Range-Doppler Coupling Effect and Its Suppression

In the case of conventional stepped-frequency (SF) synthetic wideband radar system, $i_{mn}\Delta f_{d}=n\Delta f$ and $f_{m}=f_{0}$; in Equation (3), Δf is the frequency interval between adjacent pulses and NΔf is the synthetic bandwidth. Each frame has the same basic frequency $f_{0}$ and the same stepped-frequency hopping. According to Equation (3), in any frame-*m*, the change of signal phase between pulses mainly depends on the phase term 2π(2R/c)Δfn, which is only related to range-R. Therefore, FFT processing or pulse compression processing in each frame can be used to obtain the distribution of scatters in the range direction; i.e., target range profile [20]. The range resolution determined by DFT is c/(2NΔf). The second-order phase term $2\pi \left(2\Delta fvT/c\right)n^{2}$ in Equation (3) may cause energy diffusion of the scattering center in range profile, but the diffusion can be ignored because the synthetic bandwidth NΔf is far smaller than the carrier frequency *f*_0_ and the phase variation of the second-order phase term is very small. The other velocity-related phase terms $2\pi \left(2f_{0}vT/c\right)n$ and 2π(2ΔfvNT/c)mn are linear with *n*, and their influence on pulse compression is that the position of the scatter on the FFT spectrum is shifted by an offset of $f_{0}v\left(1+mN\right)T/\Delta f$, which is called range-Doppler coupling effect. The range-Doppler coupling effect in an SF radar system can cause error in range measurement, but cannot cause significant diffusion of energy or degradation of imaging quality. After pulse compression and envelope alignment of a range profile in each frame, the inter-frame phase change of each range unit mainly depends on the phase term $2\pi \left(2f_{0}vNT/c\right)m$. Therefore, FFT processing or Doppler processing in each range unit can be used to obtain the distribution of scatters in velocity or Doppler direction. The distribution of the scatters on the 2D range-Doppler plane can be obtained by synthesizing the distributions of all the range units. As described above, in the conventional SF system, the imaging processing method of pulse compression in each frame at first, and then Doppler processing in each range resolution unit, are generally adopted.

It can be seen from Equation (3) that the phase of the signal is complexly related to the range and speed of the scatter due to HPRF and RFH. In each frame-*m*, the range-related phase term $2\pi \left(2R/c\right)\Delta f_{d}i_{mn}$, changes randomly and nonlinearly between pulses because *i_mn_* changes randomly. In order to use traditional FFT for pulse compression in each frame, it is necessary to rearrange the data in order of frequency from small to large, and then interpolate the non-uniform frequency-sampled data into uniform frequency-sampled data. The randomly-changed phase term $2\pi \left(2R/c\right)\Delta f_{d}i_{mn}$ is transformed to linearly-changed phase term 2π(2R/c)Δfn after rearrangement and interpolation. However, data rearrangement randomizes the original linear range-Doppler coupling phase term $2\pi \left(2f_{m}vT/c\right)n$. For supersonic/hypersonic applications, even if the clutter-center velocity compensation is made by using Equation (2), the phase change of the coupling phase term caused by the velocity residual *v* is still large for the scatters that are not at the direction of beam-center, and it can be close to or even more than 2*π* in one frame. The random change of phase is equivalent to adding multiplicative noise to the signal, and it seriously reduces the coherence of the rearranged data, which leads to serious energy-divergence of scatters and degradation of imaging quality. For inter-frame Doppler processing, because *i_mn_* changes randomly, the velocity-related phase term $\left(2f_{m}vNT/c\right)m$ changes randomly and nonlinearly between frames, which makes the Doppler processing complicated. The random and nonlinear range-Doppler coupling effect not only exists within the frame but occurs between frames, so it is difficult to carry out fractal-dimension processing in range and Doppler directions respectively.

In this paper, the above phenomenon is called the special range-Doppler coupling effect of RFH synthetic wideband radar in a highly dynamic application. Because of the above special effect, the conventional imaging processing method of pulse compression in each frame at first and then Doppler processing in each range resolution unit cannot be adopted in HPRF RFH radar. In order to suppress the special range-Doppler coupling effect, Doppler processing must be advanced to each frame and be synchronous with the pulse compression processing, which is called Doppler pre-processing in this paper.

The imaging algorithm of Doppler pre-processing is based on the 2D GMF algorithm, which can be executed by means of pipeline-parallel processing, and the computation can be dispersed to each frame in combination with the data acquisition process. The algorithm can be optimized in real-time ability according to different RFH modes.

As an example, the basic principle of suppressing the above special coupling effect through Doppler pre-processing is illustrated by the following intra-frame pseudo RFH mode where $i_{mn}\Delta f_{d}=i_{n}\Delta f$ and $f_{m}=f_{0}$ (the basic frequency of pulse signal remains unchanged between frames; for different *n*, $i_{n}$ randomly takes different values in [0,1,2,…,N−1] without repetition); then,
$$\begin{array}{c} y\left(n|m,k\right)=Aexp\{j2\pi [\left(2R/c\right)\Delta fi_{n}-\left(2f_{0}vNT/c\right)m-\left(2f_{0}vT/c\right)n-\\ \left(2\Delta fvT/c\right)ni_{n}-\left(2\Delta fvNT/c\right)mi_{n}+2f_{0}R/c]\}exp\left\{j\varphi \left(f_{0},R,v\right)\right\}. \end{array}$$

Obviously, for any fixed period number *n*, the phase terms $2\pi \left(2f_{0}vNT/c\right)m$ and $\left(2\Delta fvNT/c\right)mi_{n}$ vary non-randomly and linearly with the frame number *m*. Therefore, the *M* sampled data $\left\{y_{S}\left(n|m,k\right)|m=0,1,\ldots ,M-1\right\}$ of the same period number *n* and the same sampling unit number *k* can be processed first by using FFT (Doppler pre-processing). The velocity resolution converted from the DFT spectral resolution is $\Delta v=c/\left(2f_{0}NMT\right)$, and the distribution of the scatters in the velocity direction or the Doppler direction can be obtained. In the Doppler pre-processed data $\left\{Y_{S}\left(n|l_{v},k\right)|l_{v}=0,1,\ldots ,M-1\right\}$, the phase terms of the signal on the $l_{v}$-th speed channel changes with *n* are mainly $2\pi \left(2R/c\right)\Delta fi_{n}$ and $2\pi \left(2f_{0}vT/c\right)n$. Due to the accumulation or filtering effect of DFT, the change range of *v* in this channel is $\left[l_{v}\Delta v-\Delta v/2,l_{v}\Delta v+\Delta v/2\right]$. If the signal on the $l_{v}$-th speed channel is phase-compensated by the phase factor $exp\left\{j2\pi \left(2f_{0}l_{v}\Delta vT/c\right)n\right\}$ during or after Doppler processing, the phase term $2\pi \left(2f_{0}vT/c\right)n$ becomes $2\pi \left(2f_{0}v^{\prime }T/c\right)n$, where $-\Delta v/2<v^{\prime }<\Delta v/2$. As long as the accumulation time *MNT* is long enough or the resolution of velocity is high enough, Δ*v* is small enough, and the change range of the term $2\pi \left(2f_{0}v^{\prime }T/c\right)n$ can be far less than *π*/4. Rearrange the data $\left\{Y_{S}\left(n|l_{v},k\right)|n=0,1,\ldots ,N-1\right\}$ on each speed channel $l_{v}$ in the order of *i_n_* from small to large; then, the phase term $2\pi \left(2R/c\right)\Delta fi_{n}$ becomes 2π(2R/c)Δfn, while $2\pi \left(2f_{0}v^{\prime }T/c\right)n$ becomes a random phase term of small value, which can be ignored.

By FFT processing of the rearranged data on each speed channel, the distribution of the scatters along range direction of each speed channel can be obtained. By synthesizing all the speed channels, the distribution of the scatters on the 2D range-Doppler plane can be obtained. It has almost the same imaging effect as the conventional imaging algorithm used in SF Radar.

For other RFH modes, Doppler pre-processing can also suppress the above special range-Doppler coupling effect, and different fast 2D range-Doppler imaging algorithms can be obtained.

### 3.2. Image Processing Based on 2D GMF

According to the theory of matched filtering, for any transmitting signal waveform, as long as it has a certain bandwidth and a certain time width, the 2D range-Doppler image of the detected area can be obtained from the return signal through 2D matched filtering processing at the receiving end. The range resolution of the image depends on the effective bandwidth of the transmitting signal, and the speed resolution depends on the effective time-width of the signal. If the random frequencies are uniformly-distributed, the effective bandwidth is proportional to the synthetic bandwidth ΔF.

Supposing the required non-ambiguous range depth of imaging at each sampling unit is $R_{p}$, the parameters ΔF and N are designed to satisfy $R_{p}=cN/\left(2\Delta F\right)$. If range depth of $R_{p}$ is divided into N range cells, the corresponding range width of each cell is c/(2ΔF), which is exactly the nominal range resolution corresponding to the synthetic bandwidth ΔF of RFH signal. The non-ambiguous velocity measurement range $\left[0,c/\left(2f_{0}MT\right)\right]$ is divided into M velocity cells, and the velocity width corresponding to each cell is $c/\left(2f_{0}MNT\right)$, which is exactly the velocity resolution corresponding to the accumulation time *MNT* of a complex frame. According to Equation (3) and the principle of 2D GMF, the processing of 2D range-Doppler segment imaging at each sampling unit *k* can be described as follows
(4)$$\begin{array}{c} P\left(l_{u},l_{v},k\right)=\sum_{m=0}^{M-1}\sum_{n=0}^{N-1}W_{\Omega }\left(m,n\right)y_{S}\left(n|m,k\right)exp\left\{-j2\pi \left[\left(f_{m}+\Delta f_{d}i_{mn}\right)k\tau \right]\right\}\times \\ exp\left\{-j2\pi \left[\frac{f_{m}+\Delta f_{d}i_{mn}}{\Delta F}l_{u}\right]\right\}\times exp\left\{j2\pi \left[\frac{f_{m}+\Delta f_{d}i_{mn}}{f_{0}}\left(l_{v}-\frac{M}{2}\right)\frac{m}{M}\right]\right\}\times exp\left\{j2\pi \left[\frac{f_{m}+\Delta f_{d}i_{mn}}{f_{0}}\left(l_{v}-\frac{M}{2}\right)\frac{n}{MN}\right]\right\}, \end{array}$$
where $\left\{P\left(l_{u},l_{v},k\right)|l_{u}=0,1,\ldots ,N-1;l_{v}=0,1,\ldots ,M-1\right\}$ is called the 2D segment image of the target area obtained by the sampling point *k*. $P\left(l_{u},l_{v},k\right)$ is the value (complex number) at the pixel point $\left(l_{u},l_{v}\right)$, where $l_{u}$ is the number of pixel points in the range direction; $l_{v}$ is the number of pixel points in the speed direction. The total number of pixels in the segment image is *MN*.

The function of phase term $exp\left\{-j2\pi \left[\left(f_{m}+\Delta f_{d}i_{mn}\right)k\tau \right]\right\}$ is to calibrate the segment image, so that the starting position $l_{u}=0$ and the ending position $l_{u}=N-1$ of the segment image in the range direction correspond to the starting position ckτ/2 and the ending position $ck\tau /2+\left(N-1\right)R_{p}/N$. The purpose of calibration is to ensure that the segment image acquired at different sampling units is not ambiguous. $exp\left\{-j2\pi \left[\left(f_{m}+\Delta f_{d}i_{mn}\right)l_{u}/\Delta F\right]\right\}$ is the frequency-domain, non-uniform-sampling Fourier transform factor in the direction of range. The range-domain sampling after transformation is uniform, and the sampling interval is c/(2Δf), but the frequency-domain sampling interval defined by $i_{mn}\Delta f_{d}$ before transformation is non-uniform and random. $exp\left\{j2\pi \left(f_{m}+\Delta f_{d}i_{mn}\right)\left(l_{v}-M/2\right)m/\left(Mf_{0}\right)\right\}$ is the time-domain, non-uniform sampling Fourier transform factor in the velocity direction. The equivalent time-domain sampling before transformation is non-uniform because of variation of between frame, and the Doppler- frequency-domain sampling or velocity sampling after the transformation is uniform, with an interval of $c/\left(2f_{0}MNT\right)$. $exp\left\{j2\pi \left[\left(f_{m}+\Delta f_{d}i_{mn}\right)\left(l_{v}-M/2\right)n/MNf_{0}\right]\right\}$ is the range-Doppler coupling compensation factor, which can realize the phase compensation of the range movement within the sampling unit and across the sampling unit. Obviously, the motion compensation and 2D Fourier transform are carried out synchronously, and different phase compensation is used in different velocity channels to improve the compensation accuracy.

Obviously, when $i_{mn}\Delta f_{d}=n\Delta f$, $f_{m}=f_{0}$, the RFH synthesis wideband system degenerates to the conventional SF synthesis wideband system, and the 2D GMF of Equation (4) degenerates to the conventional 2D windowed DFT operation, which can be implemented by 2D fast Fourier transform.

However, the computation complexity of 2D GMF is much higher than that of 2D FFT, so it is necessary to combine different RFH modes and use fast algorithms to realize 2D GMF to meet the real-time needs of high-speed platform-borne application.

Since the range width of echoes in each sampling unit is cτ/2, if the non-ambiguous range depth of the segment image is $R_{p}$, it is required that $R_{p}\geq c\tau /2+R_{I}$, so that the echoes of scatters which move across sampling unit can be accumulated in-phase at the same point of panoramic image, and this is of importance in supersonic/hypersonic applications. *R_I_* is the maximum moving range of the scatters in an imaging period of MNT. The overlapping width of the segment image of adjacent sampling units in the range direction is $R_{p}-c\tau /2$, and the number of non-overlapping range resolution cells is $K_{d}=Nc\tau /\left(2R_{p}\right)$. Then, the panoramic image in the beam irradiation area can be obtained from the segment image of all the sampling units as follows:$$Z\left(i,j\right)=\overset{K-1}{\underset{k=0}{\sum }}P\left(i-kK_{d},j,k\right)U\left(i-kK_{d}\right)\;i=0,1,\ldots ,K_{d}+\left(K-1\right)\left(N-K_{d}\right)-1;j=0,1,\ldots ,M-1,$$
where *U*(*i*) is a rectangular function with length *N*, defined as:$$U\left(i\right)=\{\begin{array}{c} 1,\;0\leq i\leq N-1\\ 0,\;otherwise \end{array}$$

In summary, the procedure of imaging process is shown in Figure 2.

### 3.3. The Tradeoff between Randomness and Real-Time Performance

For the conventional SF radar, there is fast imaging algorithm of 2D FFT because of uniform sampling in both frequency-domain and slow-time-domain. Theoretically speaking, for the RFH radar system, the fast imaging algorithm depends on the structural characteristics of the RFH pattern. Because there are too many RFH patterns, it is impossible to design optimal imaging algorithm that has the least amount of computation for every RFH pattern. However, it is possible to design pipeline-parallel processing real-time imaging for different RFH modes combined with the data acquisition process. 

In this paper, several RFH modes are defined as follows.

#### 3.3.1. Intra-Complex-Frame Pure-RFH

In the *m*-th frame and the *n*-th pulse period of a complex frame, the frequency $f_{mn}$ of the transmitting signal is randomly selected according to certain algorithm in the frequency band $\left(f_{m},f_{m}+\Delta F\right)$. The basic frequency $f_{m}$ can hop randomly in a wide frequency range between frames. In this mode of RFH, $f_{mn}=f_{m}+\Delta f_{d}i_{mn}$, where $i_{mn}$ is the sequence number corresponding to the carrier frequency of the *m*-th frame and the *n*-th period. $i_{mn}$ is randomly and un-repeatedly selected according to a certain probability density distribution in the integer set [0,1,2,…,I−1], where *I* = Δ*F*/Δ*f_d_*. Different frame-*m* adopts different baseband frequency point set {$\Delta f_{d}i_{mn}|n=0,1,\ldots ,N-1\}$. This mode has the best performance of randomness, low-interception, and anti-interference.

#### 3.3.2. INTRA-Frame Pure-RFH

In each frame of a complex frame, the same frequency point set and the same hopping-order are adopted. For any frame *m*_1_ and *m*_2_, $f_{m_{1}n}=f_{m_{2}n}=f_{n}^{\prime }$, where $f_{n}^{\prime }$ is randomly selected according to certain algorithm in the frequency band $\left(f_{0},f_{0}+\Delta F\right)$. Because the frequency is the same between frames, the initial phase *φ_mn_* of the transmitting signal must be randomly selected according to certain algorithm between 0 and *π*, so as to reduce the cyclic autocorrelation of the transmitting signal and maintain the low interception performance. For different complex frames, the basic frequency $f_{0}$ randomly changes in a large range as much as possible. The same frequency between frames can simplify the Doppler processing and improve the real-time performance. However, compared with the intra-complex-frame pure-RFH mode, this mode loses performance of randomness, low-interception, and anti-interference because the same frequency point set and the same hopping-order are adopted in each frame.

In this mode of RFH, $f_{m}=f_{0}$, $i_{mn}=i_{n}$ and $f_{mn}=f_{0}+\Delta f_{d}i_{n}$, where $i_{n}$ is randomly and un-repeatedly selected according to a certain probability density distribution in the range of integer set [0,1,2,…,I−1], where *I* = Δ*F*/Δ*f_d_*.

#### 3.3.3. Intra-Complex-Frames Pseudo-RFH

The baseband frequency points are obtained by uniform sampling in (0,ΔF), $f_{mn}=f_{m}+i_{mn}^{\prime }\Delta F/N$, where n=0,1,2,…,N−1. The basic frequency *f_m_* can hop randomly in a wide frequency range between frames, and $i_{mn}^{\prime }$ can be randomly selected according to certain algorithm in the range of {0,1,2,…,N−1}. In this mode, different frames can adopt different basic frequencies and different hopping-orders but the same uniformly sampled baseband frequency point set, and the pulse compression processing in the range direction can be done by fast Fourier transform after higher order motion compensation and data rearrangement, which improves the real-time performance. Compared with the above two modes, the shortcoming of this mode is that the non-ambiguous range depth of segment image at each sampling unit decreases because of the frequency-domain uniform sampling. It is necessary to increase *N* and reduce the frequency hopping interval Δf=ΔF/N to meet the design requirements of non-ambiguous range depth. In addition, this mode loses more performance of randomness, low-interception, and anti-interference.

In this mode, $\Delta f_{d}i_{mn}=\left(\Delta F/N\right)i_{mn}^{\prime }$ and $f_{mn}=f_{m}+\left(\Delta F/N\right)i_{mn}^{\prime }$, where $i_{mn}^{\prime }$ is randomly and un-repeatedly selected according to a certain probability density distribution in the range of integer set [0,1,2,…,N−1].

#### 3.3.4. Intra-Frame Pseudo-RFH

The baseband frequency points are obtained by uniform sampling in (0,ΔF), $f_{mn}=f_{0}+i_{n}^{\prime }\Delta F/N$, where n=0,1,2,…,N−1. The basic frequency does not hop between frames. $i_{n}^{\prime }$ is randomly and un-repeatedly selected according to certain algorithm in the range of {0,1,2,…,N−1}. The initial phase *φ_mn_* is randomly selected according to certain algorithm between 0 and *π*. For different complex frames, the basic frequency $f_{0}$ randomly changes in a large range as much as possible. In this mode, both the pulse compression processing in the range direction and the Doppler processing in the speed direction can be done by fast Fourier transform, which further improves the real-time performance. Decreasing in non-ambiguous range depth, and more loss in performance of randomness, low-interception, and anti-interference, are also the shortcomings of this mode.

See Appendix A for the specific generation of 2D RFH patterns for those four RFH modes

### 3.4. Online, Fast 2D Imaging Algorithms for Different RFH Modes

As mentioned before, in order to avoid image defocusing caused by the special range-Doppler coupling in the case of RFH, Doppler pre-processing must be carried out synchronized with the pulse compression process. However, Doppler processing is a kind of inter-frame processing. It will cause a serious delay in signal processing if Doppler processing is not done until the data of the last frame is collected. Considering that the data of the RFH synthetic wideband radar is obtained in the order of frames and periods, pipeline-parallel processing can be used to divide the Doppler pre-processing and pulse compression into each frame and each period, which can reduce the delay in signal processing.

#### 3.4.1. Intra-Complex-Frame Pseudo-RFH Mode

In the 2D matched filtering (range-Doppler imaging) equation of Equation (4), in each frame of each sampling unit, the data are rearranged in the way of frequency point from small to large. Set $n^{\prime }$ as the frequency point number after rearrangement and the corresponding number before rearrangement is *n_m_*. Set the rearranged data as $y_{S}^{\prime }\left(n^{\prime }|m,k\right)$. According to the definition of this RFH mode $\Delta f_{d}i_{mn^{\prime }}=n^{\prime }\Delta F/N$, so the 2D matched filtering of Equation (4) can be re-written as follows:(5)$$\begin{array}{r} P\left(l_{u},l_{v},k\right)=\sum_{m=0}^{M-1}\sum_{n=0}^{N-1}W_{\Omega }\left(m,n_{m}\right)y_{S}^{\prime }\left(n^{\prime }|m,k\right)\times exp\left\{-j2\pi \left[\left(f_{m}+n^{\prime }\Delta F/N\right)k\tau \right]\right\}\\ \times exp\left\{-j2\pi n^{\prime }l_{u}/N\right\}\times exp\left\{j2\pi \left[\frac{f_{m}+n^{\prime }\Delta F/N}{f_{0}}\left(l_{v}-\frac{M}{2}\right)\frac{m}{M}\right]\right\}\\ \times exp\left\{-j2\pi \frac{f_{m}}{\Delta F}l_{u}\right\}\times exp\left\{j2\pi \left[\frac{f_{m}+n^{\prime }\Delta F/N}{f_{0}}\left(l_{v}-\frac{M}{2}\right)\frac{n_{m}}{MN}\right]\right\} \end{array}$$

Imaging Algorithm 1: FFT-based pulse-compression on multiple velocity channels
Step 1. Initializing, set $\left\{P^{\left(-1\right)}\left(l_{u},l_{v},k\right)=0|l_{u}=0,1,\ldots ,N-1;l_{v}=0,1,\ldots ,M-1\right\}$Step 2. For m=0,1,…,M−1, perform the following iteration:
(A)Obtain the data of the *k*-th sampling unit of the *m*-th frame: $\left\{y_{S}\left(n|m,k\right)|n=0,1,\ldots ,N-1\right\}$.(B)Data rearrangement: $\left\{y_{S}\left(n|m,k\right)|n=0,1,\ldots ,N-1\right\}\to \left\{y_{S}^{\prime }\left(n^{\prime }|m,k\right)|n^{\prime }=0,1,\ldots ,N-1\right\}$.(C)Windowing, range calibration, multi-velocity-channel motion compensation, and Doppler pre-processing:(6)$$y_{S}^{\prime \prime }\left(n^{\prime }|m,k,l_{v}\right)=W_{\Omega }\left(m,n_{m}\right)y_{S}^{\prime }\left(n^{\prime }|m,k\right)\psi \left(m,n^{\prime },k,l_{v}\right),$$
where
(7)$$\begin{array}{c} \psi \left(m,n^{\prime },k,l_{v}\right)=exp\{-j2\pi \left[\left(f_{m}+n^{\prime }\Delta F/N\right)k\tau \right]+j2\pi \left(f_{m}+n^{\prime }\Delta F/N\right)\left(l_{v}-\frac{M}{2}\right)m/\left(f_{0}M\right)+\\ j2\pi \left(f_{m}+n^{\prime }\Delta F/N\right)\left(l_{v}-\frac{M}{2}\right)n_{m}/\left(f_{0}MN\right)\}. \end{array}$$(D)Multi-velocity-channel fast pulse-compression processing.It can be obtained according to Equations (5) and (6) that
$$P\left(l_{u},l_{v},k\right)=\sum_{m=0}^{M-1}exp\left\{-j2\pi \frac{f_{m}}{\Delta F}l_{u}\right\}\sum_{n^{\prime }=0}^{N-1}y_{S}^{\prime \prime }\left(n^{\prime }|m,k,l_{v}\right)\times exp\left\{-j2\pi n^{\prime }l_{u}/N\right\}.$$The operation $\sum_{n^{\prime }=0}^{N-1}y_{S}^{\prime \prime }\left(n^{\prime }|m,k,l_{v}\right)\times exp\left\{-j2\pi n^{\prime }l_{u}/N\right\}$ is a uniform sampling DFT, which can be realized by *FFT*:(8)$$\left\{Y_{S}^{\prime \prime }\left(l_{u},l_{v}|m,k\right):l_{u}=0,1,\ldots ,N-1\right\}=FFT\left\{y_{S}^{\prime \prime }\left(n^{\prime }|m,k,l_{v}\right):n^{\prime }=0,1,\ldots ,N-1\right\}.$$(E)Current-frame Doppler-accumulation processing:(9)$$P^{\left(m\right)}\left(l_{u},l_{v},k\right)=P^{\left(m-1\right)}\left(l_{u},l_{v},k\right)+exp\left\{-j2\pi \frac{f_{m}}{\Delta F}l_{u}\right\}\times Y_{S}^{\prime \prime }\left(l_{u},l_{v}|m,k\right)\;\left(l_{u}=0,1,\ldots ,N-1;l_{v}=0,1,\ldots ,M-1\right).$$

Obviously, $P^{\left(M-1\right)}\left(l_{u},l_{v},k\right)=P\left(l_{u},l_{v},k\right)$.

In this algorithm, gradually, a clear image is obtained through iteration. Every additional frame of data increases the sharpness of the image. Because the rearranged frequency points are uniformly sampled, the pulse compression processing on each velocity-channel can be realized by FFT, which improves the real-time performance of the imaging algorithm.

#### 3.4.2. Intra-Frame Pseudo-RFH Mode

This RFH mode is equivalent to making all $f_{m}=f_{0}$ in the intra-complex-frame pseudo-RFH mode. After data rearrangement, the *n_m_* corresponding to $n^{\prime }$ is the same, which is labeled as *n* and independent of *m*. In this mode, there is neither coupling phase term of *l_u_* and *m*, nor coupling phase term *l_v_* and *n*. Compared to imaging algorithm 1, the computational complexity can be further reduced by using fractal-dimension processing.

Imaging Algorithm 2. Fractal-dimension processing with Doppler pre-processing
Step 1. Data rearrangement. Rearrange the data $y_{S}\left(n|m,k\right)$ of each frame in the order of frequency points from small to large. The rearranged data are $y_{S}^{\prime }\left(n^{\prime }|m,k\right)$, in which the number before rearrangement corresponding to $n^{\prime }$ is *n*.Step 2. Windowing and velocity calibration ($l_{v}=M/2$ corresponds to zero-velocity after compensation)
(10)$$y_{S}^{\prime \prime }\left(n^{\prime }|m,k\right)=W_{\Omega }\left(m,n_{m}\right)y_{S}^{\prime }\left(n^{\prime }|m,k\right)\times exp\left\{-j2\pi \left[\frac{f_{0}+n^{\prime }\Delta F/N}{f_{0}}\frac{m}{2}\right]\right\}\;\left(m=0,1,\ldots ,M-1\right).$$Step 3. For the data with the same sampling unit number *k*, pulse period number $n^{\prime }$, and different frame number *m*, carry out the non-integer sampling (*l_v_* is an integer, but $\left(f_{0}+n^{\prime }\Delta F/N\right)l_{v}/f_{0}$ is not an integer) IDFT processing:(11)$$Y_{S}^{\prime \prime }\left(l_{v}|n^{\prime },k\right)=\overset{M-1}{\underset{m=0}{\sum }}y_{S}^{\prime \prime }\left(n^{\prime }|m,k\right)\times exp\left\{j2\pi \left[\frac{f_{0}+n^{\prime }\Delta F/N}{f_{0}}l_{v}\frac{m}{M}\right]\right\}\;\left(l_{v}=0,1,\ldots ,M-1\right).$$Step 4. Range calibration and multi-velocity-channel motion compensation.
(12)$$\begin{array}{c} Y_{S}^{\prime }\left(l_{v}|n^{\prime },k\right)=Y_{S}^{\prime \prime }\left(l_{v}|n^{\prime },k\right)\\ \times exp\left\{-j2\pi \left[\left(f_{0}+n^{\prime }\Delta F/N\right)k\tau \right]+j2\pi \left[\frac{f_{0}+n^{\prime }\Delta F/N}{f_{0}}\left(l_{v}-\frac{M}{2}\right)\frac{n}{MN}\right]\right\}. \end{array}$$Step 5. Range-dimension pulse-compression processing on each velocity channel:(13)$$P\left(l_{u},l_{v},k\right)=\overset{N-1}{\underset{n^{\prime }=0}{\sum }}Y_{S}^{\prime }\left(l_{v}|n^{\prime },k\right)\times exp\left\{-j2\pi n^{\prime }l_{u}/N\right\}\;\left(l_{u}=0,1,\ldots ,N-1\right).$$

Obviously, the DFT processing of Equation (13) can be realized by FFT.

#### 3.4.3. Intra-Frame Pure-RFH Mode

In this mode, $f_{m}=f_{0}$, $i_{mn}=i_{n}$, and the common phase factor $exp\left\{-j2\pi f_{m}l_{u}/\Delta F\right\}$ can be ignored, so Equation (4) can be written as follows:(14)$$\begin{array}{c} P\left(l_{u},l_{v},k\right)=\sum_{m=0}^{M-1}\sum_{n=0}^{N-1}W_{\Omega }\left(m,n\right)y_{S}\left(n|m,k\right)\times exp\left\{-j2\pi \left[\left(f_{0}+\Delta f_{d}i_{n}\right)k\tau \right]\right\}\times \\ exp\left\{-j2\pi \frac{\Delta f_{d}i_{n}}{\Delta F}l_{u}\right\}\times exp\left\{j2\pi \left[\frac{f_{0}+\Delta f_{d}i_{n}}{f_{0}}\left(l_{v}-\frac{M}{2}\right)\frac{m}{M}\right]\right\}\times exp\left\{j2\pi \left[\frac{f_{0}+\Delta f_{d}i_{n}}{f_{0}}\left(l_{v}-\frac{M}{2}\right)\frac{n}{MN}\right]\right\}. \end{array}$$

Imaging Algorithm 3. Pipeline-parallel processing 2D matching filtering algorithm
Step 1. Initialize, set $\left\{P^{\left(-1\right)}\left(l_{u},l_{v},k\right)=0|l_{u}=0,1,\ldots ,N-1;l_{v}=0,1,\ldots ,M-1\right\}$.Step 2. For m=0,1,…,M−1, perform the following iteration:
(A)Obtain the data of the *k*-th sampling unit of the *m*-th frame: $\left\{y_{S}\left(n|m,k\right)|n=0,1,\ldots ,N-1\right\}$;(B)Windowing, range calibration, multi-velocity-channel motion compensation, and Doppler pre-processing: (15)$$y_{S}^{\prime \prime }\left(n|m,k,l_{v}\right)=W_{\Omega }\left(m,n\right)y_{S}\left(n|m,k\right)\psi \left(m,n,k,l_{v}\right),$$
where
(16)$$\begin{array}{c} \psi \left(m,n,k,l_{v}\right)=exp\{-j2\pi \left[\left(f_{0}+\Delta f_{d}i_{n}\right)k\tau \right]+j2\pi \left(f_{0}+\Delta f_{d}i_{n}\right)\left(l_{v}-\frac{M}{2}\right)m/\left(f_{0}M\right)+\\ j2\pi \left(f_{0}+\Delta f_{d}i_{n}\right)\left(l_{v}-\frac{M}{2}\right)n/\left(f_{0}MN\right)\}. \end{array}$$(C)Multi-velocity-channel pulse-compression processingIt can be obtained according to Equations (14) and (15) that
$$P\left(l_{u},l_{v},k\right)=\sum_{m=0}^{M-1}\sum_{n=0}^{N-1}y_{S}^{\prime \prime }\left(n|m,k,l_{v}\right)\times exp\left\{-j2\pi \frac{\Delta f_{d}i_{n}}{\Delta F}l_{u}\right\}.$$The operation $Y_{S}^{\prime \prime }\left(l_{u},l_{v}|m,k\right)=\sum_{n=0}^{N-1}y_{S}^{\prime \prime }\left(n|m,k,l_{v}\right)\times exp\left\{-j2\pi \Delta f_{d}i_{n}l_{u}/\Delta F\right\}$ is non-uniform sampling DFT. Even if the data are rearranged in the order of frequency points from small to large, the rearranged data are still non-uniform samples, which are difficult to be realized by FFT before inserting into uniform samples.
(17)$$\left\{Y_{S}^{\prime \prime }\left(l_{u},l_{v}|m,k\right):l_{u}=0,1,\ldots ,N-1\right\}=NUDFT\left\{y_{S}^{\prime \prime }\left(n|m,k,l_{v}\right):n=0,1,\ldots ,N-1\right\}.$$(D)Current-frame Doppler accumulation processing
(18)$$P^{\left(m\right)}\left(l_{u},l_{v},k\right)=P^{\left(m-1\right)}\left(l_{u},l_{v},k\right)+Y_{S}^{\prime \prime }\left(l_{u},l_{v}|m,k\right)\;\left(l_{u}=0,1,\ldots ,N-1;l_{v}=0,1,\ldots ,M-1\right).$$

Imaging Algorithm 4. Multi-velocity-channel pulse-compression based on data rearrangement/interpolation/range dimension FFT

In order to ensure the accuracy of interpolation, the interpolation processing must be done in each velocity channel. Due to the strong randomness of phase $\left(2f_{0}vT/c\right)n$ after data rearrangement, the rearranged data needs to be processed by Doppler pre-processing, so that the change range of signal phase $\left(2f_{0}\Delta vT/c\right)n$ on each speed channel is far less than one, where Δv is the width of velocity resolution unit. On each velocity channel, the effect of rearranged random phase term $\left(2f_{0}\Delta vT/c\right)n$ is negligible. The data accumulated by Doppler pre-processing are rearranged and interpolated on each velocity channel, which makes it is easy to ensure the interpolation accuracy.
Step 1. Windowing and velocity calibration.
(19)$$y_{S}^{\prime \prime }\left(n|m,k\right)=W_{\Omega }\left(m,n_{m}\right)y_{S}\left(n|m,k\right)\times exp\left\{-j2\pi \left[\frac{f_{0}+\Delta f_{d}i_{n}}{f_{0}}\frac{m}{2}\right]\right\}\;m=0,1,\ldots ,M-1.$$Step 2. For the data with the same sampling unit number *k*, pulse period number *n*, and different frame number *m*, carry out the non-integer sampling (*l_v_* is an integer, but $\left(f_{0}+\Delta f_{d}i_{n}\right)l_{v}/f_{0}$ is not an integer) IDFT processing:(20)$$Y_{S}^{\prime \prime }\left(l_{v}|n,k\right)=\overset{M-1}{\underset{m=0}{\sum }}y_{S}^{\prime \prime }\left(n|m,k\right)\times exp\left\{j2\pi \left[\frac{f_{0}+\Delta f_{d}i_{n}}{f_{0}}l_{v}\frac{m}{M}\right]\right\}\;\left(l_{v}=0,1,\ldots ,M-1\right).$$Step 3. Range calibration, multi-velocity-channel motion compensation:(21)$$Y_{S}^{\prime }\left(l_{v}|n,k\right)=Y_{S}^{\prime \prime }\left(l_{v}|n,k\right)\times exp\left\{-j2\pi \left[\left(f_{0}+\Delta f_{d}i_{n}\right)k\tau \right]+j2\pi \left[\frac{f_{0}+\Delta f_{d}i_{n}}{f_{0}}\left(l_{v}-\frac{M}{2}\right)\frac{n}{MN}\right]\right\}.$$Step 4. Data rearrangement and spline interpolation. In each sampling unit and each velocity channel, the data $Y_{S}^{\prime }\left(l_{v}|n,k\right)$ is rearranged in the order of *n* corresponding frequency points from small to large to get the non-uniformly stepped-frequency sampled data. Then, the pre trained spline interpolation model is used to interpolate the non-uniformly sampled data to uniformly sampled data $Y_{S}\left(l_{v}|n^{\prime },k\right)$, and the corresponding frequency points are unified as $f_{0}$, $f_{0}+\Delta F/N$, $f_{0}+2\Delta F/N$, …, $f_{0}+\left(N-1\right)\Delta F/N$.Step 5. Perform range-dimension pulse-compression processing on each velocity channel
(22)$$P\left(l_{u},l_{v},k\right)=\overset{N-1}{\underset{n^{\prime }=0}{\sum }}Y_{S}\left(l_{v}|n^{\prime },k\right)\times exp\left\{-j2\pi n^{\prime }l_{u}/N\right\}\;\left(l_{u}=0,1,\ldots ,N-1\right).$$

Obviously, the DFT processing of Equation (22) can be realized by *FFT*.

#### 3.4.4. Intra-Complex Frame Pure-RFH Mode

Imaging Algorithm 5. Pipeline-parallel processing 2D matched filtering algorithm
Step 1. Initialize, set $\left\{P^{\left(-1\right)}\left(l_{u},l_{v},k\right)=0|l_{u}=0,1,\ldots ,N-1;l_{v}=0,1,\ldots ,M-1\right\}$.Step 2. For m=0,1,…,M−1, perform the following iteration:
(A)Obtain the data of the *k*-th sampling unit of the *m*-th frame: $\left\{y_{S}\left(n|m,k\right)|n=0,1,\ldots ,N-1\right\}$;(B)Windowing, range calibration, multi-speed channel motion compensation, and pre-processing of Doppler: (23)$$y_{S}^{\prime \prime }\left(n|m,k,l_{v}\right)=W_{\Omega }\left(m,n\right)y_{S}\left(n|m,k\right)\psi \left(m,n,k,l_{v}\right),$$
where
(24)$$\begin{array}{c} \psi \left(m,n,k,l_{v}\right)=exp\{-j2\pi \left[\left(f_{0}+\Delta f_{d}i_{mn}\right)k\tau \right]+j2\pi \left(f_{0}+\Delta f_{d}i_{mn}\right)\left(l_{v}-\frac{M}{2}\right)m/\left(f_{0}M\right)+\\ j2\pi \left(f_{0}+\Delta f_{d}i_{mn}\right)\left(l_{v}-\frac{M}{2}\right)n/\left(f_{0}MN\right)\}. \end{array}$$(C)Multi-velocity-channel pulse-compression processing.It can be obtained according to Equations (4) and (23) that
$$P\left(l_{u},l_{v},k\right)=\sum_{m=0}^{M-1}exp\left\{-j2\pi \frac{f_{m}}{\Delta F}l_{u}\right\}\sum_{n=0}^{N-1}y_{S}^{\prime \prime }\left(n|m,k,l_{v}\right)\times exp\left\{-j2\pi \frac{\Delta f_{d}i_{mn}}{\Delta F}l_{u}\right\}.$$The operation $\sum_{n=0}^{N-1}y_{S}^{\prime \prime }\left(n|m,k,l_{v}\right)\times exp\left\{-j2\pi \Delta f_{d}i_{mn}l_{u}/\Delta F\right\}$ is non-uniform sampling DFT:(25)$$\left\{Y_{S}^{\prime \prime }\left(l_{u},l_{v}|m,k\right):l_{u}=0,1,\ldots ,N-1\right\}=NUDFT\left\{y_{S}^{\prime \prime }\left(n|m,k,l_{v}\right):n=0,1,\ldots ,N-1\right\}.$$(D)Current-frame Doppler accumulation processing
(26)$$P^{\left(m\right)}\left(l_{u},l_{v},k\right)=P^{\left(m-1\right)}\left(l_{u},l_{v},k\right)+exp\left\{-j2\pi \frac{f_{m}}{\Delta F}l_{u}\right\}\times Y_{S}^{\prime \prime }\left(l_{u},l_{v}|m,k\right)\;\left(l_{u}=0,1,\ldots ,N-1;l_{v}=0,1,\ldots ,M-1\right).$$

Obviously, $P^{\left(M-1\right)}\left(l_{u},l_{v},k\right)=P\left(l_{u},l_{v},k\right)$.

### 3.5. Real-Time 1D Hig-Range-Resolution (HRR) Imaging Algorithm of HPRF RFH Radar

The 1D HRR imaging algorithm is mainly used in the stage of target-tracking. As mentioned before, due to the special range-Doppler coupling effect of RFH synthetic wideband imaging radar in supersonic/hypersonic applications, the HRR imaging processing algorithm is essentially different from that of conventional stepped frequency synthetic wideband imaging radar, which can obtain HRR range profile by using the data of one frame. In RFH radar, if we want to obtain the range profile of the current frame, Doppler pre-processing should be executed before the current frame in order to suppress the special-range Doppler coupling effect.

The 1D HRR imaging algorithm is the same as the above online, fast, 2D imaging algorithms. The difference is that, in the target tracking stage, the target has been detected by the range-Doppler 2D image acquired in the searching stage, and the sampling unit and velocity channel where the target is located have been measured; then, the range-Doppler imaging processing algorithm only needs to be carried out on the velocity channel of the target and its adjacent channel. The multi-velocity-channel range-Doppler 2D imaging and splicing processing only need to be carried out in the corresponding sampling unit and the adjacent sampling units. That can significantly reduce the complexity of computation. In order to meet the requirements of high data rate in the tracking phase, the multi-velocity-channel range profile must be updated by frame, not by complex-frame as in 2D imaging. It is easy to design the iterative algorithm that can obtain the range profile of the next frame from that of the current frame by adding a few computations.

## 4. Experimental Results and Evaluation of Fast Imaging Algorithm

The diving angle of the aircraft was *θ_M_* = −30° (the angle between the moving direction of the radar platform and the horizontal plane), the flight speed was *V_M_* = 1750 m/s, and the height was *H* = 30 km. The pulse repetition period was *T* = 14 us, the carrier basic frequency was *f*_0_ = 35 GHz, the frequency hopping interval was Δ*f* = 6.25 MHz, and the pulse width was *τ* = 0.08 us. The number of periods in one frame was *N* = 16, the corresponding synthetic wideband was *N*Δ*f* = 100 MHz, the range resolution was Δ*R* = c/(2*N*Δ*f*) = 1.5 m, the number of frames in one complex-frame was *M* = 32, the corresponding period of complex-frame was *MNT* = 7.168 ms, and the velocity resolution was Δ*v* = *c*/(2*f*_0_*MNT*) = 0.6 m/s.

We assumed that the target was a stationary ship on the sea, and was equivalent to seven strong scattering centers. The parameters of each scattering center are shown in Table 1.

According to the definition of resolution-cell in the above 2D GMF method, the theoretical coordinates of those seven scattering centers in spliced 2D range image are shown in Table 2.

Sea clutters were simulated according to average SCR of 8 dB. The intra-complex-frame pseudo-RFH mode of $i_{mn}\Delta f_{d}=i_{mn}^{\prime }\left(\Delta F/N\right)$ and $f_{m}=f_{0}$ was investigated first, and the 2D range-Doppler normalized image obtained by 2D GMF is shown as Figure 3.

The other RFH modes were also used in the simulation experiments, and the imaging results were almost the same except the side-lobe level. The side-lobe level of pure-RFH mode was higher than that of pseudo-RFH mode. The simulation results show that the 2D GMF method can obtain high quality images in all modes of RFH. However, the computation-complexity of this method is much higher than that of 2D FFT. The 2D GMF was implemented by different pipeline-parallel algorithm according to different RFH mode in order to improve the real-time performance.

For Imaging Algorithm 1, because the rearranged data are uniformly sampled in frequency domain, the pulse-compression processing on each velocity channel can be realized by FFT, which improves the real-time performance of the imaging algorithm. The 2D GMF needs $2\times {\left(MN\right)}^{2}$ complex multiplication operations, but the algorithm of FFT-based pulse-compression on multiple velocity channels needs only $M^{2}\left[3N+N\log\left(N\right)\right]$ complex multiplication operations. The total operation is reduced to [log(N)+3]/N times of the original.

Imaging Algorithm 2 needs MN[M+log(N)+3] complex multiplication operations. Compared with Algorithm 1, the computation is much less, but it is not convenient for pipeline processing, and the delay time of signal processing is not necessarily short.

Imaging Algorithm 3 needs $M^{2}N^{2}+2M^{2}N$ complex multiplication operations, which is more than for Imaging Algorithm 1 and Imaging Algorithm 2. However, the operations can be decomposed to each frame for execution, and the delay time of signal processing is short. It can meet the requirements of real-time imaging by multiprocessor parallel processing, and each processor is responsible for windowing, motion compensation, pulse compression, and Doppler accumulation of several velocity channels.

Regardless of the operations of low-order spline interpolation, Imaging Algorithm 4 needs MN[M+log(N)+3] complex multiplication operations, which is equivalent to that of Algorithm 2. It is also inconvenient for pipeline processing, and the delay time of signal processing is not necessarily shorter than that of Algorithm 3.

Imaging Algorithm 5 needs $M^{2}N^{2}+3M^{2}N$ complex multiplication operations. Similar to Imaging Algorithm 3, the operations can be decomposed to each frame for execution, and the delay time of signal processing is short. It can meet the requirements of real-time imaging processing through multiprocessor parallel processing.

The imaging results of the five imaging algorithms are almost the same as that of the 2D GMF. Figure 4 shows the 2D normalized image obtained by Imaging Algorithm 1.

For the intra-frame pseudo-RFH mode, as a contrast, Figure 5 shows the imaging results obtained by the traditional fractal-dimension imaging algorithm without Doppler pre-processing. In this method, the data $y_{S}\left(n|m,k\right)$ are re-arranged in the order of frequency points from small to large in each frame, and a new data sequence $y_{S}^{\prime }\left(n|m,k\right)$ is obtained. Then, the range calibration is performed by multiplying the phase factor $exp\left\{-j2\pi \left[\left(f_{0}+n\Delta F/N\right)k\tau \right]\right\}$, and the range profile $y_{S}^{\prime }\left(l_{u}|m,k\right)$ of each frame is obtained by processing the calibrated data with *N*-point DFT (pulse compression). Finally, DFT (Doppler beam sharpening) processing of *M*-point is carried out for data of M frames in each range resolution cell $l_{u}$, and the 2D normalized image $P\left(l_{u},l_{v},k\right)$ is obtained.

Obviously, due to the special range-Doppler coupling effect caused by RFH and the lack of Doppler pre-processing in the pulse-compression process, the image is seriously divergent.

For the conventional SF mode, as a contrast, Figure 6 shows the normalized imaging results obtained by using the traditional fractal-dimension imaging algorithm.

Compared with Figure 3 and Figure 4, it is shown that the RFH synthesis wideband system can achieve almost the same imaging effect as that of the conventional SF system, but the side-lobe level is higher in image of RFH system.

## 5. Discussion

In the case of RFH mode and a supersonic/hypersonic application, the conventional fractal-dimension 2D range-Doppler imaging algorithm makes it difficult to obtain high quality images because of the special range-Doppler coupling. Theoretical analysis and simulation results show that the proposed pipeline-parallel processing fast imaging algorithms based on Doppler pre-processing and 2D GMF can well suppress the above special range-Doppler coupling effect, avoid the divergence of the image in the range direction, and meet the requirements of real-time imaging. However, the side-lobe level of pure-RFH mode is higher than that of pseudo-RFH mode. Further, it is necessary to suppress the side-lobe level by optimizing the RFH pattern and the 2D window function $W_{\Omega }\left(m,n\right)$.

## Figures and Tables

**Figure 1 sensors-19-05424-f001:**
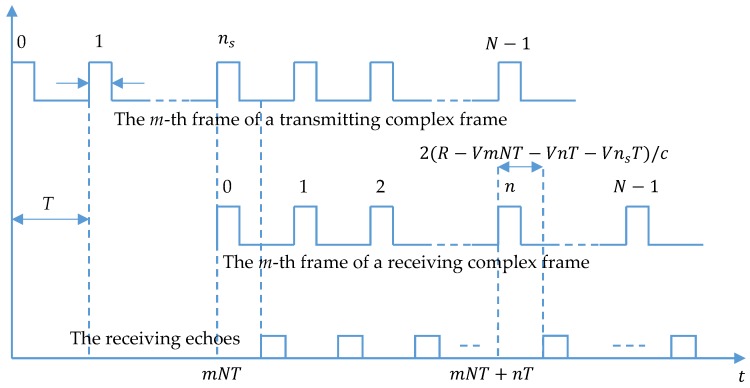
Sequence chart of each frame.

**Figure 2 sensors-19-05424-f002:**
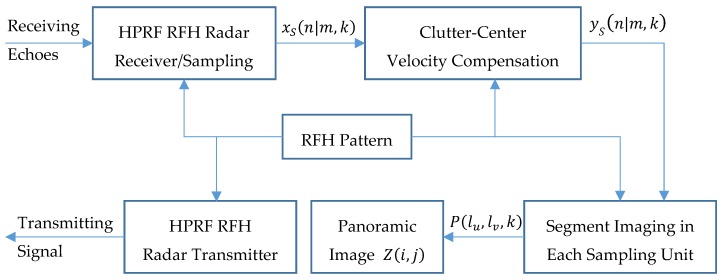
The procedure of imaging processing.

**Figure 3 sensors-19-05424-f003:**
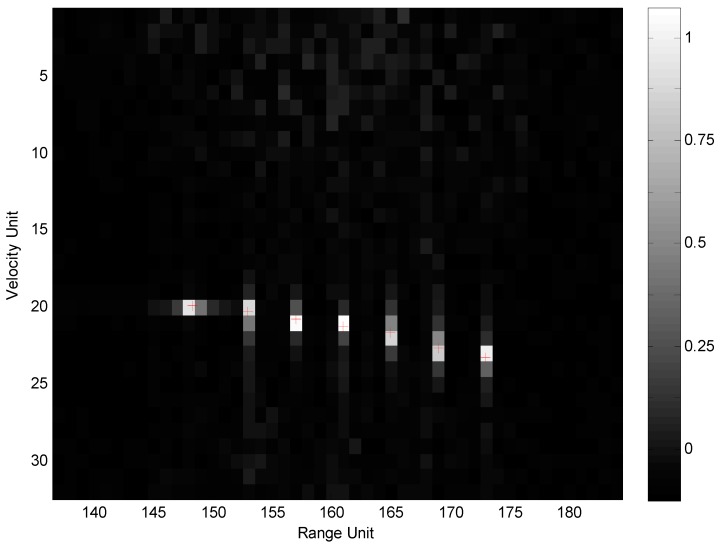
Image obtained by 2D GMF.

**Figure 4 sensors-19-05424-f004:**
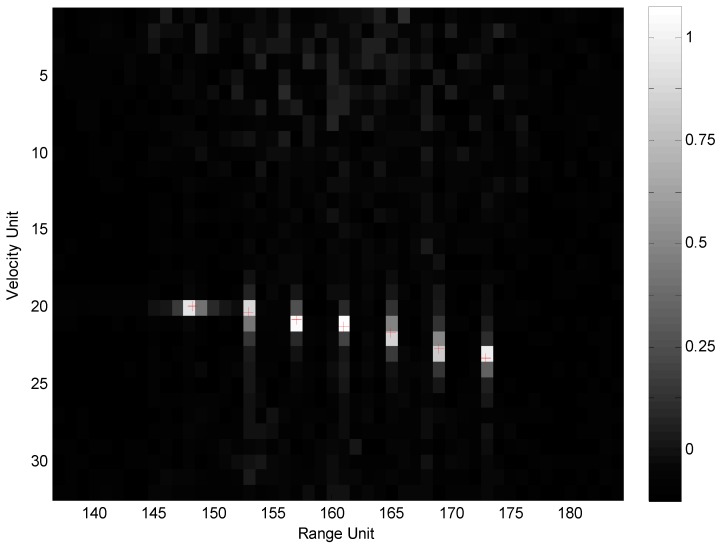
Image obtained by Imaging Algorithm 1.

**Figure 5 sensors-19-05424-f005:**
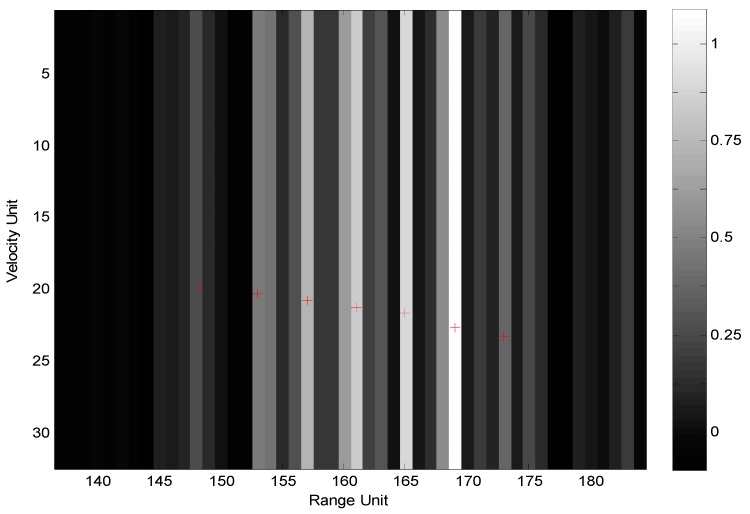
Image obtained by traditional algorithm without Doppler pre-processing.

**Figure 6 sensors-19-05424-f006:**
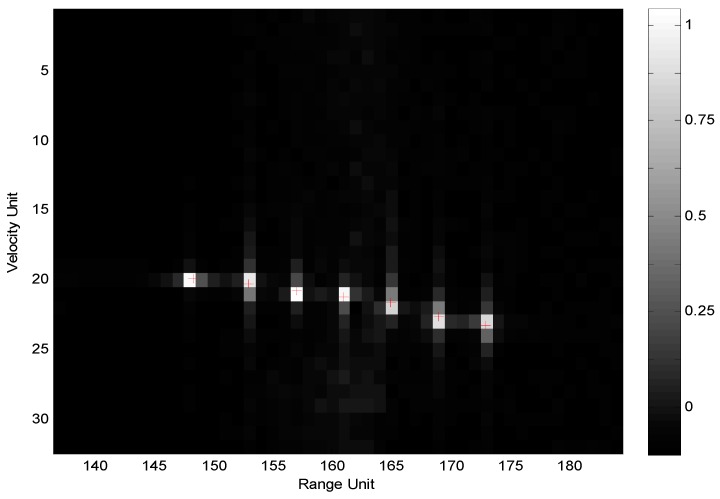
Image obtained by conventional stepped frequency (SF) mode.

**Table 1 sensors-19-05424-t001:** Parameters of scattering centers of target.

Scattering Center Number	Initial Distance/m	Pitch Angle/(°)	Azimuth /(°)	Normalized Scattering Intensity
1	33,822	−59.883	10	1
2	33,828	−59.866	10	1
3	33,834	−59.848	10	1
4	33,840	−59.831	10	1
5	33,846	−59.814	10	1
6	33,852	−59.774	10	1
7	33,858	−59.747	10	1

**Table 2 sensors-19-05424-t002:** Theoretical positions of each scattering center of the target in image.

Scattering Center Number	Range Cell	Velocity Cell
1	148	20
2	153	20
3	157	21
4	161	21
5	165	22
6	169	23
7	173	23

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
