# Peer review of "Research of a Radar Imaging Algorithm Based on High Pulse Repetition Random Frequency Hopping Synthetic Wideband Waveform"

_sensors, 2019, doi:10.3390/s19245424_

Round 1
Reviewer 1 Report
Please see file attached

Author Response
We would like to thank the reviewer for his careful read and thoughtful comments on previous draft. We have carefully taken the comments into consideration in preparing our revision.
Please see the attachment.
Thanks for all the help.

Reviewer 2 Report
This paper proposes an imaging method for the special case of radar systems. The proposal seems reasonable however; the descriptions are too complicated and difficult to understand.
1. Both [1] and [2] are someone's thesis. Is there any open and widely read article for the topic?
2. For Section 2 and 3, please use figures for describing the signal processing, geometric coordinate and flowchart. It is hardly understandable.
3. In L. 274, characters are overlapped.
4. Fig. 1, 2 and 4 are quite similar to each other. Please show the difference further and if possible, show their oversampled images of the point targets to evaluate the focusing results.
Author Response

(The authors gave the same response as above.)

Reviewer 3 Report
General impression of the paper: An original solution for imaging of high speed targets.
It is recommended authors to cite the following paper where a physical interpretation of Fast Fourier Transform is discussed.
A. Lazarov, Ch. Minchev. ISAR Geometry, Signal model and image processing algorithms, IET Journal Radar, Sonar and Navigation, vol. 11, No 9, Sept. 2017, p. 1425 – 1434, DOI: 10.1049/iet-rsn.2017.0081.
The authors have to answer all reviewer's comments in an attached file.

Author Response

(The authors gave the same response as above.)

Round 2
Reviewer 2 Report
The revised paper seems mostly fine except for that the power scale is absent for Figs. 3-6.
Author Response
Point 1: The revised paper seems mostly fine except for that the power scale is absent for Figs. 3-6.
Response 1:Thank you for the advice. We added colormaps in Figs 3-6, and we used intensity normalization (normalization with the maximum value of amplitude). The minimum value is zero, and the maximum value is 1. We also made the corresponding explanation in this paper.